# Mild Cardiotoxicity and Continued Trastuzumab Treatment in the Context of HER2-Positive Breast Cancer

**DOI:** 10.3390/jcm12216708

**Published:** 2023-10-24

**Authors:** Orianne de la Brassinne Bonardeaux, Benjamin Born, Marie Moonen, Patrizio Lancellotti

**Affiliations:** 1Department of Cardiology, GIGA Cardiovascular, University of Liège Hospital, 4000 Liège, Belgium; 2Intensive Care Department, Citadelle of Liège Hospital, 4000 Liège, Belgium

**Keywords:** trastuzumab cardiotoxicity, cardiomyopathy, HER2-positive breast cancer, echocardiography

## Abstract

Breast cancer is the leading cause of cancer death in women worldwide. Trastuzumab, the main HER2-targeted treatment, faces limitations due to potential cardiotoxicity. The management of patients with mild cardiotoxicity on trastuzumab remains uncertain, resulting in treatment discontinuation and negative oncological outcomes. This retrospective study analyzed 23 patients who experienced decreased left ventricular function during trastuzumab treatment. During the 18-month follow-up period, two patients (9%) had severe declines in function, leading to treatment cessation, and one patient (4%) developed heart failure symptoms. However, 21 patients showed mild, reversible myocardial dysfunction without significant differences in final ventricular function compared to a control group (58.4% vs. 61.7%, respectively; *p* = 0.059). The declines in function were most pronounced at nine months but improved at twelve and eighteen months. Various echocardiographic parameters changed significantly over time. As predictors of severe cardiotoxicity, we identified the following: LVEF before initial chemotherapy (*p* = 0.022), as well as baseline LVEF before treatment with trastuzumab (*p* = 0.007); initial left ventricular end systolic volume (*p* = 0.027); and the initial global longitudinal strain (*p* = 0.021) and initial velocity time integral in the left ventricular outflow track (*p* = 0.027). In conclusion, the continuation of trastuzumab should be considered for most patients with mild cardiotoxicity, with close cardiac monitoring and cardioprotective measures. However, identifying the patients at risk of developing severe cardiotoxicity is necessary. According to our data, the initial LVEF and GLS levels appear to be reliable predictors.

## 1. Introduction

Breast cancer is the most diagnosed cancer in women and accounts for over two million new cancer cases worldwide each year [1]. The incidence rates are highest in Northern and Western Europe, North America, and Oceania, while they are lower in Asia and Sub-Saharan Africa [2]. Belgium has one of the highest incidence rates in Europe [3].

Although breast cancer mortality has been decreasing in Western countries since the 1970s due to the development of screening strategies and adjuvant treatments [4,5], breast cancer still remains the leading cause of cancer-related deaths in women worldwide. In 2018, Europe recorded 522,513 new cases of breast neoplasia and 137,707 breast cancer-related deaths [1].

During the initial evaluation of breast neoplasia, it is routinely recommended to determine the expression of estrogen receptors (ER) and progesterone receptors (PR) by tumor cells, as well as to assess the overexpression of human epidermal growth factor 2 (HER2) receptors. This information has a significant impact on both the prognosis and treatment decisions [6,7].

HER2 is a transmembrane glycoprotein belonging to the epidermal growth factor receptors (EGFR) family. It has intrinsic tyrosine kinase activity and is involved in various intracellular signaling mechanisms that control growth, survival, adhesion, migration, and cellular differentiation [8]. Approximately 15 to 20% of breast cancers overexpress the HER2 protein.

Patients with HER2-positive breast cancer often receive neoadjuvant therapies before surgery depending on locoregional extension.

Trastuzumab (TTZ), marketed as Herceptin^®^, is the most prescribed HER2-targeted treatment. It is a humanized monoclonal antibody that blocks HER2 activation by binding to its extracellular domain. This inhibits intracellular signaling, controlling cell proliferation and differentiation [9]. TTZ may also recruit immune cells for tumor cell lysis and has other suggested mechanisms, such as receptor internalization.

Studies have shown that combining TTZ with chemotherapy improves complete response rates, long-term survival, and event-free survival in HER2-positive breast cancer patients, both in neoadjuvant and adjuvant settings [10,11]. In practice, TTZ is administered intravenously or subcutaneously on a weekly basis or every three weeks for a standard duration of one year.

TTZ’s main limitation is its cardiotoxicity, typically presenting as asymptomatic decreases in left ventricular ejection fraction (LVEF) and, in rare cases, as symptomatic heart failure [12,13]. This toxicity is more likely to occur in patients who have previously received anthracycline-based chemotherapy, which is also known for its cardiotoxicity [14].

Two types of myocardial dysfunction are associated with anticancer treatments: type I (associated with anthracyclines) that causes the irreversible destruction of myocytes, and type II (associated with TTZ and other treatments), which is characterized by myocyte dysfunction that does not result in cell death [15]. Type II myocardial dysfunction is not dose-dependent, does not result in ultrastructural changes allowing potential functional recovery after a few months, and, in a majority of cases, the reintroduction of treatment after recovery. These two distinct types of cardiotoxicities can coexist in the same patients in cases of sequential chemotherapy.

In patients treated with conventional chemotherapy and TTZ, the incidence of symptomatic heart failure ranges from 0.8% to 3.3% compared to 0.45% in patients treated with chemotherapy alone [16,17,18]. Asymptomatic reductions in LVEF have been observed in 2.4% to 7.2% of cases, although some studies have reported incidences as high as 27% [17,19,20]. TTZ toxicity typically occurs within the treatment year and not beyond [21]. 

In the event of myocardial dysfunction, it may be necessary to temporarily or even permanently discontinue TTZ. Based on different studies [22], clinicians have been advised to interrupt its administration in cases of decreases in LVEF of ≥15% or of ≥10% below the lower limit of normal, a fall in LVEF of below 40%, or the onset of heart failure symptoms. TTZ may be continued when LVEF remains above 40%, provided there is close cardiac monitoring and appropriate cardioprotective medication [23]. Myocardial dysfunction secondary to TTZ generally responds well to conventional treatment for systolic heart failure. This involves initiating treatment with beta-blockers and angiotensin-converting enzyme inhibitors (ACEI) or angiotensin receptor blockers (ARBs) [23].

Due to limited data, there is no consensus on the management of patients experiencing asymptomatic and moderate deterioration of LVEF while on TTZ treatment. 

Recently, two studies have demonstrated the relative safety of continuing TTZ despite mild cardiotoxicity. Leong et al. followed 20 patients with slightly impaired LVEF on TTZ and found that 90% of the subjects did not experience any cardiac complications during the treatment year, while 10% of patients developed severe heart failure requiring definitive treatment discontinuation [24]. In another retrospective study by Barron et al., 18 patients with mildly impaired LVEF did not develop major cardiac events related to TTZ when closely monitored [25]. Based on these findings, at the University Hospital of Liège, clinicians have been continuing TTZ treatment in certain patients with mild and asymptomatic myocardial dysfunction since 2015, provided they receive close cardiac monitoring and appropriate cardioprotective treatment. However, patients with symptomatic heart failure or LVEF measurements of less than or equal to 40% have their treatment discontinued, in alignment with the current recommendations.

The objectives of our study were as follows:To evaluate the outcomes (i.e., mortality, hospitalizations, development of signs or symptoms of heart failure, and impairment in left ventricular systolic function) in patients who develop mild cardiotoxicity under TTZ when the treatment is continued.To monitor the changes in echocardiographic parameters during treatment and up to six months after the discontinuation of TTZ.To investigate the potential predictive factors for the development of severe cardiotoxicity.

## 2. Materials and Methods

This clinical study was retrospective, monocentric, and analyzed the demographic, clinical, echocardiographic, and therapeutic data, as well as the clinical and echocardiographic outcomes, of patients who developed cardiotoxicity during TTZ treatment.

The studied population included 23 female subjects who developed decreases in left ventricular systolic function during treatment with TTZ that was initiated between 1 January 2015 and 30 September 2019. TTZ treatment was administered for HER2-positive breast neoplasia from stage I to IV. HER2-positive status was defined by the overexpression of the HER2 protein on immunohistochemistry (IHC 3+) or the overexpression of the HER2 gene on fluorescence in situ hybridization (FISH) for patients with IHC 2+. Impaired LVEF was defined as a value strictly less than 54% (corresponding to the lower limit of the normal range for women) with a decrease in LVEF greater than or equal to 10% (≥10%) compared to the baseline reference. The exclusion criteria were a history of hospitalization for heart failure in the year prior to TTZ introduction or the need to interrupt TTZ treatment for a reason other than cardiotoxicity (i.e., allergy, intolerance, etc.). A control group, consisting of 21 patients treated with TTZ for HER2-positive breast neoplasia who did not experience cardiotoxicity during treatment, was also selected for secondary subgroup analysis.

### 2.1. Follow-Up Modalities

Each patient undergoing TTZ treatment within our institution benefited from quarterly cardiac monitoring, including at least one cardiac ultrasound throughout the 12-month duration of TTZ therapy. In cases of cardiotoxicity, follow-ups were intensified with cardiology consultations every 3 to 6 weeks for clinical evaluation, echocardiography, and potential therapeutic adjustments. After completing TTZ treatment, follow-up appointments were provided every 6 months unless otherwise indicated.

Furthermore, ongoing communication was established among the various specialists involved in the management of the patients (i.e., oncologists, radiation oncologists, surgeons, radiologists, pathologists, and cardiologists) before, during, and after oncological treatment. In this study, data were collected over the 12-month period of TTZ treatment and the subsequent 6 months.

### 2.2. Echocardiographic Data

A comprehensive transthoracic echocardiogram (using a General Electric Vivid E95 from GE General Healthcare, Chicago, IL, USA) was performed prior to any chemotherapy and TTZ treatment. It was repeated at least quarterly during the year of TTZ treatment and semi-annually the following year. The images were recorded, and the following values were measured and documented in the protocol: LVEF, LV longitudinal function (i.e., MAPSE, mitral annular plane systolic excursion, septal and lateral, and septal and lateral mitral s’ wave), left ventricular dimensions and volumes, diastolic function, filling pressures (i.e., E wave, A wave, and mean E/e’ ratio), left atrial volume, valvulopathy and its grading (from 1 to 4), right ventricular dimensions and longitudinal function (i.e., TAPSE, tricuspid annular plane systolic excursion, and tricuspid s’ wave), transtricuspid gradient, and the time-velocity integral within the LV outflow tract (LVOT VTI). Only two operators, using recorded ultrasound images from our institution’s server, validated the echocardiographic parameters included in this study to minimize interpretation biases.

### 2.3. Data Collection

Baseline and follow-up variables (i.e., demographic, clinical, and therapeutic) were extracted from medical records and entered into a database. Echocardiographic parameters were measured using the recorded images in the computerized database. Each death and major cardiac event were reported. Major cardiac events were defined as cardiovascular-related death, hospitalization due to heart failure, symptomatic heart failure (i.e., NYHA class III or IV dyspnea, orthopnea, edema, or unexplained weight gain), or the asymptomatic deterioration of LVEF below 40%.

### 2.4. Statistical Analysis

Categorical variables were presented as numbers and percentages. Continuous variables were reported as means and standard deviations. One-way analysis of variance (ANOVA) was performed to assess the evolution of the variables over the 18-month follow-up period. Cubic spline was fitted for variables that showed significant changes during the follow-up period, with a significance level (α) of 0.05 and confidence interval for the fit. Post-ANOVA comparisons were conducted using the Dunnett test, with the “baseline” characteristics as the control group. Matching of the controls was performed using hierarchical clustering in JMP Pro 14 (SAS Institute, Cary, NC, USA) with the following five randomization factors: body mass index (BMI), age, LVEF before TTZ treatment, hypertension, and diabetes. We used *t*-tests or Fisher’s exact tests for post-matching verification and for comparing the subgroups of patients with mild and severe cardiotoxicity.

All tests were conducted bilaterally, and a *p*-value of <0.05 was considered statistically significant. All statistical analyses were performed using SAS 9.4 unless otherwise specified (SAS Institute, Cary, NC, USA).

## 3. Results

### 3.1. Baseline Characteristics of the Study Population

From 1 January 2015 to 30 September 2019, 168 HER2-positive breast cancer patients received TTZ at the University Hospital of Liège. Among them, 24 patients (14%) met the inclusion criteria for cardiotoxicity (Figure 1). One patient discontinued treatment due to intolerance, and no patients were hospitalized for heart failure in the previous year. Hence, our study included 23 patients. The baseline characteristics of the study population are summarized in Table 1.

### 3.2. Evolution of Cardioprotective Treatment

The utilization rates of the ACEI, ARBs, and beta-blockers are presented in Table 2. These values demonstrated high rates of prescriptions for ACEI and ARBs (48% and 26% at 18 months, respectively), as well as for beta-blockers (61% at 18 months). The prescriptions for cardioprotective treatments increased during the follow-ups with the patients, while the mean arterial pressures and resting HRs decreased.

### 3.3. Clinical Evolution of the Studied Population

During the 18-month follow-up period, no deaths or hospitalizations for heart failure were recorded. Two patients (9%) experienced significant impairments in left ventricular systolic function (LVEF of <40%) and required treatment discontinuation, and one of them (4%) exhibited symptoms of heart failure classified as NYHA IV dyspnea. 

Additionally, we wanted to determine if the 21 patients who did not experience any major cardiac events, reported as mild cardiotoxicity, experienced any residual impacts on their left ventricular systolic function. For this purpose, we randomly selected a control group of 21 patients matched to the first group in terms of age, risk factors, comorbidities, and baseline LVEF (Table 3). The mean LVEF at 18 months in the mild cardiotoxicity group was 58.4%. The mean LVEF at 18 months in the control group was 61.7%. There were no significant differences in the LVEF values at 18 months between the patients with mild cardiotoxicity and the control group (*p* = 0.059), albeit with small sample sizes.

### 3.4. Evolution of Echocardiographic Parameters

We observed significant variations in the LVEF values over time, with an average LVEF decreasing from 61.6 ± 4.1% at baseline to 55.1 ± 5.3% at 3 months (*p*-value vs. baseline = 0.001), 53.9 ± 5.1% at 6 months (*p*-value vs. baseline < 0.001), 51.9 ± 6.3% at 9 months (*p*-value vs. baseline < 0.001), 54.4 ± 5.3% at 12 months (*p*-value vs. baseline < 0.001), and 57.2 ± 6.5% at 18 months (*p*-value vs. baseline = 0.042). The evolution of the LVEF values is represented in Figure 2. The results showed progressive decreases in LVEF values that occurred rapidly after the initiation of TTZ. These declines in LVEF continued and reached a nadir at 9 months of treatment. Subsequently, there were improvements in LVEF at 12 and 18 months, although there were marked differences between the two subgroups (the average LVEF values at 18 months were 58.4 ± 5.3% for the 21 patients with mild cardiotoxicity and 46 ± 7.1% for the 2 patients with severe cardiotoxicity).

In addition to the LVEF values, we observed significant variations in other echocardiographic parameters during follow-up compared to the baseline values (Figure 3 and Table 4). Anova tests were completed for every variable during the follow-up period of 18 months. Using the baseline values as the control group, post hoc multiple comparisons were performed using Dunnett’s tests. The results are presented in Table 5.

Regarding the LVESV values, we observed a mirror-like evolution compared to the LVEF measurements, with a peak at 9 months (*p* = 0.002) followed by decreases at 12 and 18 months. The septal and lateral mitral annular systolic velocities exhibited decreases from baseline to 9 months (*p* = 0.007 and 0.005, respectively), followed by increases at 12 and 18 months. The A wave velocities followed the same pattern. The septal MAPSE decreased until the twelfth month and partially recovered thereafter. The other studied parameters did not show significant changes compared to the baseline values for the study population (Appendix A).

### 3.5. Identification of Predictive Variables for Severe Cardiotoxicity

Low LVEF values before initial chemotherapy and before starting TTZ treatment pre- disposed patients to severe cardiotoxicity (Figure 4). The same trend was observed for higher initial LVESV values (*p* = 0.027), lower absolute GLS values (*p* = 0.027), and lower initial LVOT VTI values (*p* = 0.021). Finally, it appeared that no demographic or clinical variables were predictive of developing severe cardiotoxicity.

## 4. Discussion

### 4.1. Cardiotoxicity and Clinical Outcome

Our study observed a cardiotoxicity rate of 14%, which was slightly higher than the approximate 10% reported in other studies [22,24]. This could be attributed to our population’s characteristics, including its multiple risk factors such as age over 50, prior anthracycline treatment (70%), arterial hypertension (43%), and being overweight. Additionally, variations in the criteria for defining cardiotoxicity contributed to the differences across studies [26,27,28]. We found a 9% rate of severe cardiotoxicity, consistent with the literature, particularly the study by Leong et al., which reported a 10% severe toxicity rate. This highlighted two distinct subgroups of patients with cardiotoxicity [26,27,28]. Most patients (approximately 90%) experienced moderate decreases in LVEF, remained asymptomatic, and completed treatment without major complications. LV systolic function significantly recovered in the medium term, with no significant sequelae. However, a minority (approximately 10%) of patients developed more significant myocardial dysfunction, which was sometimes symptomatic, leading to discontinued TTZ and less notable recovery. Identifying these patients early was crucial for minimizing treatment effects and long-term sequelae [26,27,28]. The HFA-ICOS risk assessment tools, while needing further validation, could be an efficient screening tool [26,29,30].

Regarding the evolution of cardioprotective treatments, we observed significant increases in the prescriptions for ACEI, ARBs, and beta-blockers over time, indicating careful cardiac monitoring and high adherence to treatment recommendations for myocardial dysfunction [26,27,28]. This cardioprotective treatment likely limited the occurrence of cardiac events and hindered the deterioration of systolic function. Close cardiac monitoring remains essential in the management of patients undergoing TTZ treatment [26,27,28].

Additionally, it is important to acknowledge the evolving landscape of breast cancer therapeutics, notably, the emergence of HER2-low breast cancer defined as a score of 1+ on IHC analysis or as an IHC score of 2+ and negative results on in situ hybridization [31,32,33]. More and more studies are proving the superiority of antibody–drug conjugates such as trastuzumab deruxtecan (TTZ combined with a topoisomerase 1 inhibitor) over second or third line chemotherapy regimens [31,32,33]. The most concerning side effects shown were interstitial lung disease and pleuresia [34]. Further studies could concentrate on cardiotoxicity as TTZ is involved in this treatment [14]. Another notable treatment is pertuzumab as is it often combined with TTZ. In this case, studies have shown that the cardiotoxicity of the combined antibodies targeting HER2 is similar to TTZ alone, ranging from 7 to 12% depending on the trial [35,36,37]. However, due to limited cardiac safety data and the frequent co-administration of pertuzumab with anthracyclines and/or trastuzumab, the label carries a warning for decreased LVEF [35].

### 4.2. Impact of TTZ Therapy on Cardiac Function

Impaired LVEF, reflected by the decreases in values, occurred early after the introduction of TTZ therapy, as differences were already observed at 3 months. However, this dysfunction appeared to be transient as LVEF subsequently recovered, characteristic of type II toxicity explained by the absence of ultrastructural changes in myocytes. Interestingly, LV systolic function improved starting from the ninth month of treatment, potentially due to the introduction of treatment with ACEI, ARBs, or beta-blockers 3 to 6 months earlier. Our study identified additional echocardiographic parameters for detecting and monitoring cardiotoxicity. The LV ESV evolved in a mirror image of LVEF, which was easily understandable since they are related through the following formula: LVEF = (EDV − ESV)/EDV. The end diastolic volume (EDV) did not vary significantly during follow-up. Tissue Doppler imaging had already demonstrated its usefulness in evaluating the toxicity of anticancer treatments, including TTZ [26,27,28]. In our study, the velocities of the mitral s waves evolved in parallel with LVEF and served as reliable indicators of systolic dysfunction. We also observed transient decreases in the velocities of the mitral A waves during treatment, potentially due to decreases in atrial contractility (similar to the ventricles), elevations in LV tele-diastolic pressures, or a combination of both mechanisms. Other studies have already suggested the development of secondary diastolic dysfunction caused by TTZ [26,27,28].

It is noteworthy that our study did not find significant changes in the average global longitudinal strain (GLS), which contradicted existing evidence suggesting its reliability in detecting early myocardial toxicity during cancer treatment [26,27,28]. This discrepancy could be attributed to a lack of initial data due to technical limitations in strain measurement and to the heterogeneity of the study population [38]. Indeed, the baseline GLS values of the patients in the severe subgroup were already lower compared to those in the mild cardiotoxicity subgroup (−16.7 ± 0.9% vs. −20.3 ± 1.5%, respectively). Among those with severe impairment, detailed analysis revealed that both patients had prior cardiac histories. One patient had a history of heart failure due to hypertensive heart disease, while the other had peripartum cardiomyopathy resulting in persistent dilated cardiomyopathy, and both had preserved EF at the start of this study. Of note, an initial LVEF at the lower limit of normal was not only a risk factor but also a predictor of severe impairment. This could be explained by the compromise in the contractile reserve either due to the toxicity of prior chemotherapy or as a result of underlying cardiac pathology. A low initial GLS also reflected reduced myocardial reserve, thereby predisposing patients to more severe impairment during TTZ administration. Interestingly, a recent study [39] showed that, in rats, donepezil (an acetylcholinesterase inhibitor) protected against trastuzumab-induced cardiotoxicity through reducing multiple programmed cell death pathways. This suggested new possible management for TTZ-induced cardio toxicity.

### 4.3. Strengths and Limitations

Few studies have explored the safety of continuing TTZ in the presence of confirmed cardiotoxicity, despite numerous publications on TTZ toxicity. Our study, with the largest cohort to date in this context, demonstrated methodological rigor and an extensive analysis of the parameters, providing a medium-term outcome assessment. This study had several limitations. First, its retrospective and monocentric nature, as well as its small sample size, call for caution in interpreting the results. Second, as identified above, there was a lack of significant changes in GLS, which contradicted existing evidence. Third, the patients did not all receive the same treatment regimen (e.g., previous anthracycline-based chemotherapy or concomitant treatment with pertuzumab), and some regimens are known for causing cardiotoxicity, which could lead to confounding variables.

## 5. Conclusions and Perspectives

With close cardiac monitoring and appropriate cardioprotective treatment, the continuation of TTZ should be considered for most patients in cases of mild cardiac toxicity, aiming to improve the treatment of HER2-positive breast cancer. However, it is necessary to identify the minority of patients at risk for developing severe cardiotoxicity. According to our results, initial EF and GLS values may serve as reliable predictive factors for severe cardiac impairment.

The existing literature is characterized by small sample sizes and inconsistent protocols because of a lack of information regarding the continuation of TTZ in patients with mild cardiotoxicity. Prospective trials with large cohorts aimed at identifying those for whom TTZ should be considered contraindicated, identifying prognostic factors (clinical, biological, or imaging) or scores such as HFA-ICOS risk scores that can identify populations at risk for developing severe cardiotoxicity or analyzing the long-term evolution of echocardiographical parameters after TTZ treatment would be useful. Lastly, to gain comprehensive insights into cardiotoxicity dynamics, a trial could compare trastuzumab-associated cardiotoxicity across distinct treatment subgroups, including varying chemotherapies (e.g., anthracyclines or taxanes), pertuzumab treatments, or antibody–drug conjugates.

## Figures and Tables

**Figure 1 jcm-12-06708-f001:**
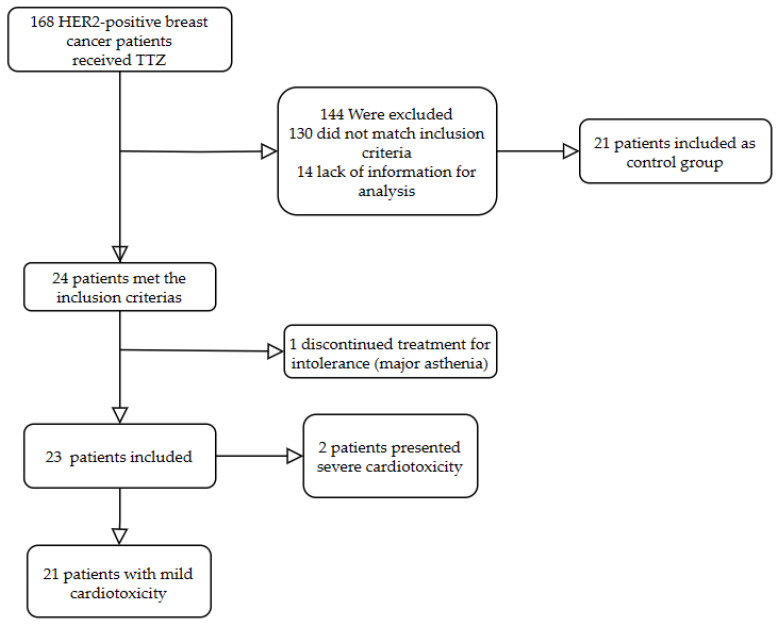
Flow chart.

**Figure 2 jcm-12-06708-f002:**
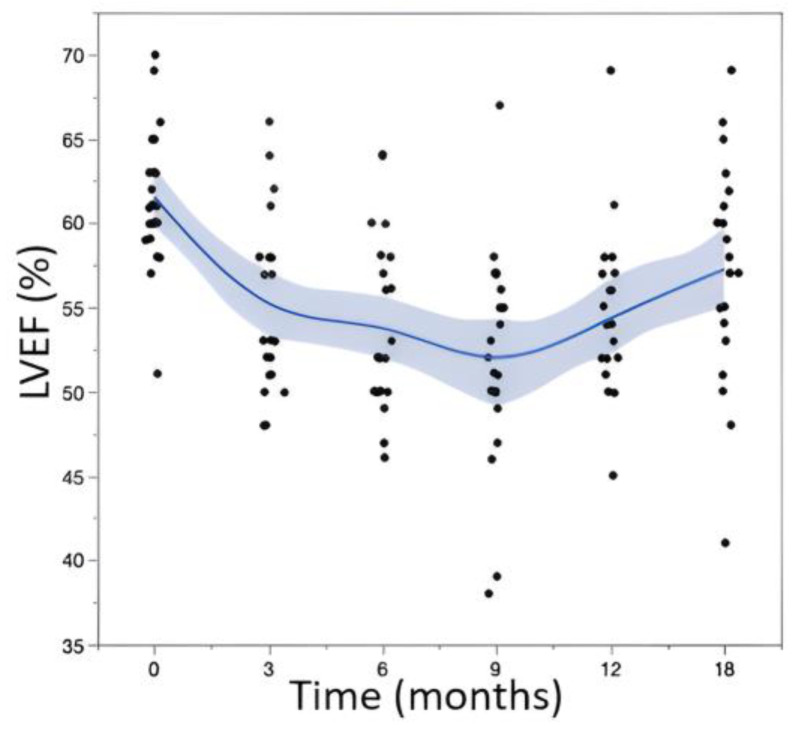
Evolution (using a fitted cubic spline) of LVEF values during follow-up. T0 corresponds to the initiation of TTZ.

**Figure 3 jcm-12-06708-f003:**
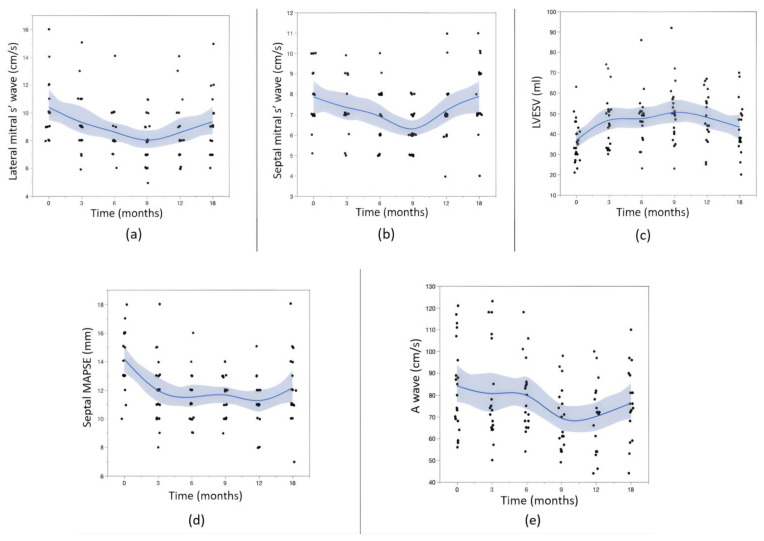
Evolution (using a fitted cubic spline) of the other echocardiographic parameters: (**a**) lateral mitral s’ wave, (**b**) septal mitral s’ wave, (**c**) LVESV (left ventricular end systolic volume), (**d**) septal MAPSE, and (**e**) A wave.

**Figure 4 jcm-12-06708-f004:**
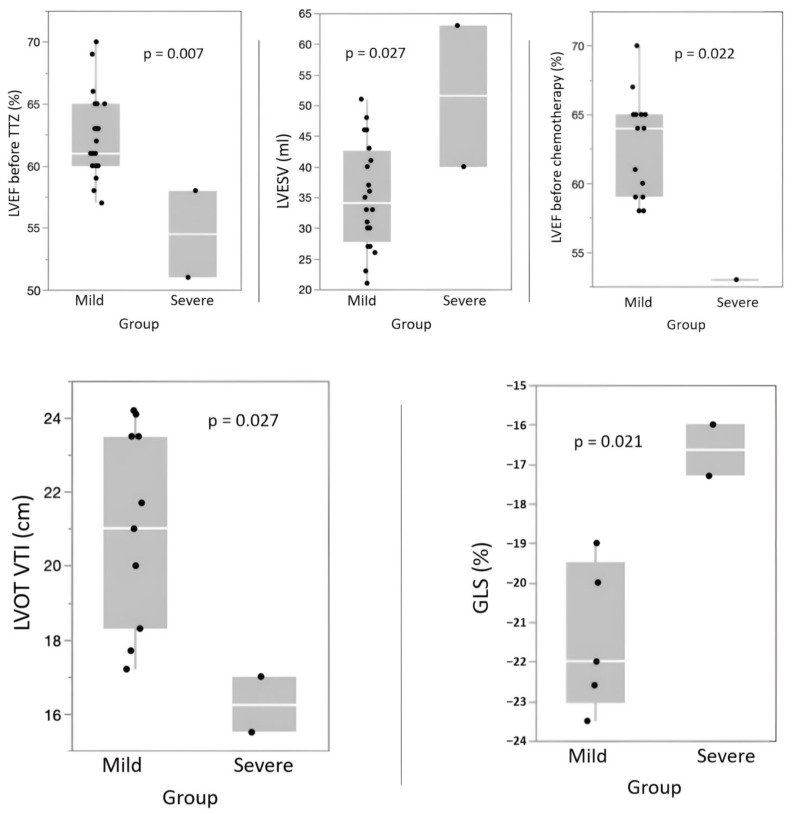
Box plots (using *t*-tests or Fisher’s exact tests for comparisons) representing the subgroups of mild cardiotoxicity versus severe cardiotoxicity.

**Table 1 jcm-12-06708-t001:** Baseline characteristics.

Demographics		Chronic Treatment	
Female gender	23 (100%)	ACEI	3 (13%)
Age (years)	56 ± 10	ARB	2 (9%)
BMI kg/m²	27.3 ± 5.8	Beta blocker	5 (22%)
Invasive ductal carcinoma	22 (96%)	Loop diuretic	1 (4%)
Undifferentiated carcinoma	1 (4%)	Clinical data	
HER-2-positive status	23 (100%)	NYHA class:	
ER-positive status	12 (52%)	– I	22 (96%)
PR-positive status	10 (43%)	– II	0 (0%)
Left laterality	9 (39%)	– III	1 (4%)
Neoplasm stage		– IV	0 (0%)
– Stage I	12 (52%)	Systolic blood pressure, mmHg	127 ± 23
– Stage II	4 (17%)	Diastolic blood pressure, mmHg	76 ± 11
– Stage III	1 (4%)	Heart rate (HR), beats per minute	79 ± 12
– Stage IV	6 (26%)	Echographical data	
Previous surgical treatment	8 (35%)	LVEF before chemotherapy, %	62.1 ± 4.3
Previous thoracic radiotherapy		LVEF before TTZ, %	61.6 ± 4.1
– Left side	1 (4%)	Left ventricular end systolic volume, ml	95 ± 24
– Right side	3 (13%)	Left ventricular end-diastolic volume, ml	37 ± 10
Previous chemotherapy		Septal mitral s’ wave, cm/s	8 ± 1
– With anthracyclines	16 (70%)	Lateral mitral s’ wave, cm/s	10 ± 2
– Without anthracyclines	1 (4%)	Septal MAPSE, mm	14 ± 2
Previous hormone therapy	0 (0%)	Lateral MAPSE, mm	15 ± 3
Previous TTZ therapy	2 (9%)	LVOT VTI, cm	20.4 ± 3
Previous pertuzumab therapy	0 (0%)	Left atrial volume, ml	49 ± 14
Concurrent pertuzumab therapy	6 (26%)	E wave, cm/s	75 ± 12
Concurrent taxane chemotherapy	23 (100%)	A wave, cm/s	84 ± 21
Risk factors and comorbidities		E/A ratio	0.9 ± 0.3
Smoking history	6 (26%)	E wave deceleration time, ms	187 ± 48
Actively smoking	3 (13%)	E/e’ mean ratio	8.5 ± 1.8
Alcoholism	3 (13%)	Right ventricle diameter in PSLAX, mm	30 ± 3
Hypertension	10 (43%)	TAPSE	23 ± 4
Dyslipidemia	11 (48%)	Tricuspid s’ wave, cm/s	13 ± 2
Diabetes mellitus	4 (17%)	Transtricuspid gradient, mmHg	23 ± 10
Coronary heart disease	0 (0%)	Mean global longitudinal strain, %	−19 ± 2
Atrial fibrillation	1 (4%)		
Anemia	15 (65%)		

The values are expressed as means ± standard deviations or as n (%). BMI, body mass index; HER2, Human Epidermal Growth Factor Receptor-2, ER, estrogen receptor; PR, progesterone receptor; ACEI, angiotensin converting enzyme inhibitor; ARB, angiotensin receptor blocker; NYHA, New York heart association; TTZ, Trastuzumab; LVEF, left ventricular ejection fraction, MAPSE, mitral annular plane systolic excursion; LVOT VTI, initial integral of time-velocity in the left ventricular outflow tract; PSLAX, parasternal long axis; TAPSE, tricuspid annular plane systolic excursion.

**Table 2 jcm-12-06708-t002:** Utilization rates of cardioprotective treatments, evaluation of blood pressure, and HR over time. BP, blood pressure; HR, heart rate.

Time	ACEI (N = 23)	ARBs (N = 23)	Beta-blockers (N = 23)	BP (Systolic/Diastolic)	HR (/min)
Baseline	3 (13%)	2 (9%)	5 (22%)	127/76 mmHg	79
3 months	3 (13%)	2 (9%)	6 (26%)	132/76 mmHg	80
6 months	8 (35%)	2 (9%)	9 (39%)	135/78 mmHg	75
9 months	10 (43%)	4 (17%)	11 (48%)	134/82 mmHg	74
12 months	13 (57%)	4 (17%)	14 (61%)	124/73 mmHg	72
18 months	11 (48%)	6 (26%)	14 (61%)	123/71 mmHg	71

The values are expressed as n (%). ACEI, angiotensin converting enzyme inhibitor; ARB, angiotensin receptor blocker; BP, blood pressure; HR, heart rate.

**Table 3 jcm-12-06708-t003:** Comparisons, using *t*-tests or Fisher’s exact tests, between the control group and the mild toxicity group.

	Control Group (N = 21)	Mild Cardiotoxicity (N = 21)	*p*-Value	Randomization Factor
Pre-trastuzumab LVEF, %	61.9 ± 3.4	62.2 ± 3.4	0.75	Yes
Age, years	59 ± 10	56 ± 10	0.36	Yes
BMI, kg/m^2^	27 ± 6	27 ± 6	0.71	Yes
Hypertension	9 (43%)	9 (43%)	1.00	Yes
Diabetes	4 (19%)	4 (19%)	1.00	Yes
Dyslipidemia	4 (19%)	9 (43%)	0.18	No
Smoking	5 (24%)	5 (24%)	1.00	No
Coronary artery disease	1 (5%)	0 (0%)	1.00	No
Atrial fibrillation	0 (0%)	1 (5%)	1.00	No
Anemia	13 (62%)	13 (62%)	1.00	No
Prior left thoracic radiotherapy	2 (10%)	1 (5%)	1.00	No
Prior anthracycline treatment	14 (67%)	15 (71%)	0.74	No
LVEF at 18 months, %	61.7 ± 5.5	58.4 ± 5.3	0.059	No

The values are expressed as means ± standard deviations or as n (%). LVEF, left ventricular ejection fraction; BMI, body mass index.

**Table 4 jcm-12-06708-t004:** Evolution of the significantly modified parameters.

Variable	Baseline	3 Months	6 Months	9 Months	12 Months	18 Months
LVEF	61.6 ± 4.1	55.1 ± 5.3	53.9 ± 5.1	51.9 ± 6.3	54.4 ± 5.3	57.2 ± 6.5
ESV	36.7 ± 10.2	47 ± 13.3	47.1 ± 12.6	50.7 ± 14.8	48 ± 12.5	43.3 ± 13.3
Lateral s’ wave	10.4 ± 2.3	9.3 ± 2.3	8.6 ± 1.7	8 ± 1.7	8.6 ± 2.2	9.4 ± 2.3
Septal s’ wave	7.9 ± 1.5	7.3 ± 1.5	6.9 ± 1.5	6.3 ± 1	7.2 ± 1.5	7.9 ± 1.7
MAPSE septal	14.2 ± 2.2	12 ± 2.4	11.5 ± 1.7	11.7 ± 1.4	11.2 ± 1.7	12.2 ± 2.5
A wave	84.3 ± 20.7	80.4 ± 22.6	80.3 ± 16.4	68.9 ± 14.4	70.3 ± 16.2	76.3 ± 17.1

The values are expressed as means ± standard deviations. LVEF, left ventricular ejection fraction; ESV, end systolic volume; MAPSE, mitral annular plane systolic excursion.

**Table 5 jcm-12-06708-t005:** Dunnett post-ANOVA tests.

Echocardiographic Parameters	*p*-Value
LVEF at 3 months	0.001
LVEF at 6 months	<0.001
LVEF at 9 months	<0.001
LVEF at 12 months	<0.001
LVEF at 18 months	0.042
ESV at 3 months	0.045
ESV at 6 months	0.037
ESV at 9 months	0.002
ESV at 12 months	0.025
Septal s’ wave at 9 months	0.007
Lateral s’ wave at 9 months	0.005
MAPSE septal at 3 months	0.005
MAPSE septal at 6 months	<0.001
MAPSE septal at 9 months	0.001
MAPSE septal at 12 months	<0.001
MAPSE septal at 18 months	0.019
A wave at 9 months	0.041

LVEF, left ventricular ejection fraction; ESV, end systolic volume; MAPSE, mitral annular plane systolic excursion.

## Data Availability

The data presented in this study are available on request from the corresponding author.

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
