# Peer review of "Mild Cardiotoxicity and Continued Trastuzumab Treatment in the Context of HER2-Positive Breast Cancer"

_jcm, 2023, doi:10.3390/jcm12216708_

Round 1

Reviewer 1 Report

It is known that myocardial dysfunction can occur with any type of anticancer treatment (more often with anthracyclines) and treatment with various classes of targeted drugs (targeted (anti-HER2) therapy), some small molecule kinase inhibitors and specific proteasome inhibitors. To date, there is evidence in the literature that asymptomatic LV dysfunction or heart failure Trastuzumab (AT against HER2) is recorded in 1.7-20.1% of cases in such patients.

The main methods for diagnosing myocardial dysfunction during antitumor treatment with trastuzumab are considered to be echocardiography, PET, MRI and assessment of the level of biomarkers (Troponin I, NUP). The 2022 ESC Guidelines on cardio-oncology development in collaboration with EHA, ESTRO and IC-OS guidelines for assessing the risk of developing LV dysfunction after the end of anticancer therapy suggest that not only serial measurements of LVEF should be taken into account to confirm LV dysfunction, but also and more sensitive methods for detecting and confirming cardiac dysfunction, including GLS and cardiac biomarkers.

As a result of this study, based on an analysis of data from 23 patients who developed cardiotoxicity during treatment with TTZ and 21 women who received TTZ for HER2-positive breast neoplasia who did not experience cardiotoxicity during treatment, the authors evaluated the clinical, echocardiographic and therapeutic data during treatment and within 6 months. after the abolition of TTZ and based on this, potential prognostic factors for the development of severe cardiotoxicity were identified.

Strengths of the study: a wide range of studied ECHO CG parameters, frequency (every 3 months) assessment of clinical and ECHO parameters with an assessment of the characteristics of the treatment; determination of additional echocardiographic parameters to detect and monitor cardiotoxicity (mitral S wave velocity, transient decrease in mitral wave velocity A).

Weaknesses of the study: a short follow-up period after PCT (6 months), in fact, does not correspond to the period of early chronic (within 12 months after PCT) cardiotoxicity; small sample of patients; a certain contradiction of the data obtained with the results of other scientists (no significant changes were found in the average global longitudinal strain - GLS, indicating its reliability in detecting early myocardial toxicity during cancer treatment) with an unsatisfactory explanation of possible causes.

Major comments:

1. In HER2-positive breast cancer patients, the HFA-ICOS risk score is known to be used to predict cardiotoxicity, although it has moderate efficacy associated with cancer therapy, why did you not evaluate this parameter?

2. The authors did not initially plan or for some reason did not provide data on the assessment of biomarker levels (for example, hsTnI, NTproBNP, sST2) and their relationship with instrumental indicators, which would undoubtedly improve the quality of the article.

The comments are aimed at improving the quality of the study and interest in the manuscript: 

1. there is not enough information in the introduction about the types of development of cardiac dysfunction against the background of anticancer therapy - reversible and irreversible, more typical for the use of TTZ;

2. The manuscript presents the frequency of prescribing cardioprotective drugs by observation points, which makes it convenient to perceive information, but there is no such presentation of EchoCG data, perhaps not all indicators, but showing the most significant dynamics;

3. The findings are supported by the results, but they are not obviously new, in particular, in relation to the fact that the initial EF and GLS can serve as reliable predictors of severe heart failure.

Author Response

Strengths of the study: a wide range of studied ECHO CG parameters, frequency (every 3 months) assessment of clinical and ECHO parameters with an assessment of the characteristics of the treatment; determination of additional echocardiographic parameters to detect and monitor cardiotoxicity (mitral S wave velocity, transient decrease in mitral wave velocity A).

Weaknesses of the study: a short follow-up period after PCT (6 months), in fact, does not correspond to the period of early chronic (within 12 months after PCT) cardiotoxicity; small sample of patients; a certain contradiction of the data obtained with the results of other scientists (no significant changes were found in the average global longitudinal strain - GLS, indicating its reliability in detecting early myocardial toxicity during cancer treatment) with an unsatisfactory explanation of possible causes.

Major comments:

  1. In HER2-positive breast cancer patients, the HFA-ICOS risk score is known to be used to predict cardiotoxicity, although it has moderate efficacy associated with cancer therapy, why did you not evaluate this parameter?

Answer :

We thank the reviewer for his positive comments. We agree that there are some limitations related to the retrospective nature of data collection. Despite this, the present work provides important information regarding the risk of continuing treatment with chemotherapy in this type of population. We agree that the HFA-ICOS risk score may be useful in certain circumstances to assist the clinician in making management decisions. However, it should be noted that this last score still requires further validation. According to this article https://doi.org/10.3390/jcm12041278, the HFA-ICOS risk score has moderate power in predicting cancer therapy–related cardiotoxicity in HER2-positive breast cancer patients. According to the 2022 ESC cardio-oncology guidelines, this score should be considered but needs to be validated as stated in the guidelines. Moreover, this study was conducted between 2015 and 2019 when the guidelines were not available yet. This parameter would therefore be interesting to include in a further study (added as a future direction in the perspectives).

  1. The authors did not initially plan or for some reason did not provide data on the assessment of biomarker levels (for example, hsTnI, NTproBNP, sST2) and their relationship with instrumental indicators, which would undoubtedly improve the quality of the article.

Answer : We agree with the reviewer that biomarkers are now part of our armamentarium for evaluating these patients. At the time of the study, these were not part of the standards and were not reimbursed, which is still the case for the BNP in Belgium. So, the biomarkers were thus not the aim of this article but could provide interesting information in further prospective studies.

The comments are aimed at improving the quality of the study and interest in the manuscript: 

  1. there is not enough information in the introduction about the types of development of cardiac dysfunction against the background of anticancer therapy - reversible and irreversible, more typical for the use of TTZ;

Answer : We have adapted the introduction accordingly.  If the Editor agrees, we could also add the following figure and ask for permission.

Figure. Potential mechanisms of action of TTZ at the HER2 level on the surface of a cell. (Image from Clifford A. Hudis. Trastuzumab — Mechanism of Action and Use in Clinical Practice. N Engl J Med 2007;357:39-51)

  1. The manuscript presents the frequency of prescribing cardioprotective drugs by observation points, which makes it convenient to perceive information, but there is no such presentation of EchoCG data, perhaps not all indicators, but showing the most significant dynamics;

Answer : We agree with the reviewer and have provided additional information in the results section (Tables 4 and 5).

  1. The findings are supported by the results, but they are not obviously new, in particular, in relation to the fact that the initial EF and GLS can serve as reliable predictors of severe heart failure.

Answer : We thank the reviewer for the comment. It is true that the initial function assessed by LV ejection fraction or GLS does not allow us to anticipate the occurrence of cardiotoxicity. Our results confirmed this too. However, the discussion is quite different since here we examined the impact of continuing the potentially cardiotoxic treatment despite initial moderate LV dysfunction. The predictive value of these parameters under these conditions remains poorly known and will require additional studies. In this context, our study provides additional results to the limited data available in the literature by showing the rather reassuring nature of continuing treatment while insisting on close monitoring.

Reviewer 2 Report

The abstract does not reflect the results obtained (statistical significance).

The conclusions in the abstract are too far-reaching for such a sample size.

Some sentences are not supported by references, e.g. “This toxicity is more likely to occur in patients who have previously received anthracycline-based chemotherapy, also known for its cardiotoxicity.”

There are many articles on cardiotoxicity of trastuzumab, but they have not been included.

For such a low sample size, the statistical tests used are inappropriate. The presented results do not reflect reality. For this type of sample size, appropriate tests should be used, including post-hoc.

For many sentences, p-values were not indicated. “Improvement”, “decrease”, etc. - it’s not enough.

Appropriate effect size measures for the tests used were not calculated. The p-value itself is definitely not enough.

No appropriate tests were used for individual time measurements (time differences, etc.).

The results do not indicate where and what test was used.

The discussion needs complete revision. It almost only addresses 3 references.

Discussion is not a repetition of obtained results. The obtained results should be referred to references and discussed. There are many such articles.

The limitations of the manuscript are described very briefly.

Minor editing of English language required.

Author Response

The abstract does not reflect the results obtained (statistical significance).

The conclusions in the abstract are too far-reaching for such a sample size.

Answer : We thank the reviewer for the comment. Accordingly, the abstract has been improved

Some sentences are not supported by references, e.g. “This toxicity is more likely to occur in patients who have previously received anthracycline-based chemotherapy, also known for its cardiotoxicity.”

There are many articles on cardiotoxicity of trastuzumab, but they have not been included.

Answer : The introduction has been adapted and now includes more studies on cardiotoxicities of Trastuzumab.

For such a low sample size, the statistical tests used are inappropriate. The presented results do not reflect reality. For this type of sample size, appropriate tests should be used, including post-hoc. For many sentences, p-values were not indicated. “Improvement”, “decrease”, etc. - it’s not enough. Appropriate effect size measures for the tests used were not calculated. The p-value itself is definitely not enough. No appropriate tests were used for individual time measurements (time differences, etc.). The results do not indicate where and what test was used.

Answer : We thank the reviewer for these comments. The tests used were defined by our statistician in line with current recommendations and the limited sample size. Nonparametric tests were used and a 1-way ANOVA was justified for variables with serial testing (monitoring). 

We have added the most useful p-values in the text and in the table 5.

The discussion needs complete revision. It almost only addresses 3 references.

Discussion is not a repetition of obtained results. The obtained results should be referred to references and discussed. There are many such articles.

Answer : We cannot agree with this comment, especially since there is very little literature on the subject concerned. The aim is to discuss and above all to highlight the results obtained in this study. Despite this, the discussion has been revised in line with the reviewer's request.

The limitations of the manuscript are described very briefly.

Answer : Limitations were described in more detail as asked.

Reviewer 3 Report

The study addresses an important clinical question regarding the management of cardiotoxicity associated with trastuzumab treatment in HER2-positive breast cancer patients. I recommend that the authors carefully address the following points to improve the quality and readability of the manuscript.

Clarity and Organization:

The manuscript could benefit from improved organization. The introduction provides a general overview, but it might be helpful to provide a brief summary of the study's objectives and main findings at the beginning.

Consider restructuring the Materials and Methods section to provide a clearer step-by-step description of the study design, data collection, and statistical methods. This will enhance the reproducibility of the study.

Data Presentation and Interpretation:

The results section presents key findings, but it would be more informative to discuss the clinical significance of the observed changes in LVEF and other echocardiographic parameters in the context of trastuzumab treatment. Are these changes within the expected range of variability for patients receiving trastuzumab, or do they represent clinically relevant cardiotoxicity?

Discussion and Clinical Implications:

The discussion should emphasize the clinical implications of the study findings. 

Provide a more comprehensive discussion of the limitations of the study, including potential sources of bias and confounding factors. This will help readers assess the robustness of the findings.

In the conclusion section, reiterate the main findings and their clinical relevance. Also, suggest potential directions for future research in this area.

References: After the following sentence “This toxicity is more likely to occur in patients who have previously 56 received anthracycline-based chemotherapy, also known for its cardiotoxicity”, the Authors should quote a reference (e.g. Di Nardo P, et al. Chemotherapy in patients with early breast cancer: clinical overview and management of long-term side effects. Expert Opin Drug Saf. 2022 Nov;21(11):1341-1355. doi: 10.1080/14740338.2022.2151584. Epub 2022 Dec 5. PMID: 36469577.)

Minor editing of English language required

Author Response

The study addresses an important clinical question regarding the management of cardiotoxicity associated with trastuzumab treatment in HER2-positive breast cancer patients. I recommend that the authors carefully address the following points to improve the quality and readability of the manuscript.

Answer : We thank the reviewer for the positive comments

Clarity and Organization:

The manuscript could benefit from improved organization. The introduction provides a general overview, but it might be helpful to provide a brief summary of the study's objectives and main findings at the beginning.

Answer : The manuscript has been adapted accordingly.

Consider restructuring the Materials and Methods section to provide a clearer step-by-step description of the study design, data collection, and statistical methods. This will enhance the reproducibility of the study.

Answer : We agree that for more clarity a better description of the population is needed. A flow chart has been provided (Figure 2) and the population has been better described.

Data Presentation and Interpretation:

The results section presents key findings, but it would be more informative to discuss the clinical significance of the observed changes in LVEF and other echocardiographic parameters in the context of trastuzumab treatment. Are these changes within the expected range of variability for patients receiving trastuzumab, or do they represent clinically relevant cardiotoxicity?

We agree with the reviewer that the results should focus on the clinical consequences of the observed changes and their relevance to the expected variations. As described, the clinical consequence in terms of events was minimal and recovery of function was observed in the majority of cases. No deaths were observed in the moderate dysfunction group. The changes observed were probably within the expected ranges, although 10% of patients showed a major deterioration in their function. All this has been better mentioned in the amended version of the manuscript.

Discussion and Clinical Implications:

The discussion should emphasize the clinical implications of the study findings. 

Provide a more comprehensive discussion of the limitations of the study, including potential sources of bias and confounding factors. This will help readers assess the robustness of the findings.

Answer : The discussion has been improved as recommended by the reviewer and limitations have been more detailed.

In the conclusion section, reiterate the main findings and their clinical relevance. Also, suggest potential directions for future research in this area.

Answer : The manuscript has been adapted with a special emphasize on future directions

References: After the following sentence “This toxicity is more likely to occur in patients who have previously 56 received anthracycline-based chemotherapy, also known for its cardiotoxicity”, the Authors should quote a reference (e.g. Di Nardo P, et al. Chemotherapy in patients with early breast cancer: clinical overview and management of long-term side effects. Expert Opin Drug Saf. 2022 Nov;21(11):1341-1355. doi: 10.1080/14740338.2022.2151584. Epub 2022 Dec 5. PMID: 36469577.)

Answer : This was added to the manuscript, thank you for this remark.

Reviewer 4 Report

This article discusses the management of patients with mild cardiotoxicity during trastuzumab treatment for HER2-positive breast cancer. The study analyzes patients who experienced decreased left ventricular function and evaluates the outcomes and predictors of severe cardiotoxicity. The findings suggest that most patients with mild cardiotoxicity can continue trastuzumab treatment with close cardiac monitoring and cardioprotective measures. Pre-treatment ventricular function and Global Longitudinal Strain are reliable predictors of severe cardiac toxicity during trastuzumab treatment. However, the following issues are required for explaining:

1. The article should discuss the future of HER2-low era and how anti-HER2 treatment should be approached in this context. It would be beneficial to explore potential strategies for targeting HER2-low breast cancer.

2. The use of trastuzumab as a monotherapy is currently limited. It would be helpful to discuss the rationale behind this and explore potential combination therapies involving trastuzumab. For example, please mentions the occurrence of cardiotoxicity events when trastuzumab is combined with pertuzumab. Can you provide more information on the frequency and severity of these events? Are there any strategies to mitigate cardiotoxicity when using this combination therapy?

4. Are there any ongoing clinical trials investigating treatment decisions for patients who experience mild cardiotoxicity? It would be valuable to include a summary table of these trials to provide a comprehensive overview.

5. The article lacks a flowchart or diagram to illustrate the research methodology. It would be beneficial to include a visual representation of the study design and data analysis process.

6. Can you provide more details on the various echocardiographic parameters that changed significantly over time? How do these changes correlate with the occurrence and severity of cardiotoxicity?

7. Some similar studies regarding the anti-HER2 treatment and cardiotoxicity events should be cited and discussed. For example, PMID: 37561451, 37407378, 35499382, 37691124, 35562334, 37392951.

8. The authors are recommended to consider engaging a professional language editing service to ensure the clarity and coherence of the manuscript.

Moderate editing of English language required. The authors are recommended to consider engaging a professional language editing service to ensure the clarity and coherence of the manuscript.

Author Response

This article discusses the management of patients with mild cardiotoxicity during trastuzumab treatment for HER2-positive breast cancer. The study analyzes patients who experienced decreased left ventricular function and evaluates the outcomes and predictors of severe cardiotoxicity. The findings suggest that most patients with mild cardiotoxicity can continue trastuzumab treatment with close cardiac monitoring and cardioprotective measures. Pre-treatment ventricular function and Global Longitudinal Strain are reliable predictors of severe cardiac toxicity during trastuzumab treatment. However, the following issues are required for explaining:

  1. The article should discuss the future of HER2-low era and how anti-HER2 treatment should be approached in this context. It would be beneficial to explore potential strategies for targeting HER2-low breast cancer.

Answer : We thank the reviewer for the comment. However, although this was not the subject of this article, a short paragraph was added to the article in the discussion.

  1. The use of trastuzumab as a monotherapy is currently limited. It would be helpful to discuss the rationale behind this and explore potential combination therapies involving trastuzumab. For example, please mentions the occurrence of cardiotoxicity events when trastuzumab is combined with pertuzumab. Can you provide more information on the frequency and severity of these events? Are there any strategies to mitigate cardiotoxicity when using this combination therapy?

Answer : We thank the reviewed for the comment and we agree that combining drugs can increase the risk of cardiotoxicity. This was added to the discussion.

  1. Are there any ongoing clinical trials investigating treatment decisions for patients who experience mild cardiotoxicity? It would be valuable to include a summary table of these trials to provide a comprehensive overview.

Answer : Currently, there is a huge lack of literature and to our knowledge there is no ongoing study on “clinicaltrials.org”. Ongoing investigations concentrate on long term evolution, finding effective prognostic factors and efficiency of cardioprotective treatment.

  1. The article lacks a flowchart or diagram to illustrate the research methodology. It would be beneficial to include a visual representation of the study design and data analysis process.

Answer : As asked, a flow chart was added

  1. Can you provide more details on the various echocardiographic parameters that changed significantly over time? How do these changes correlate with the occurrence and severity of cardiotoxicity?

Answer : As asked, a table with the data was added in the results section

  1. Some similar studies regarding the anti-HER2 treatment and cardiotoxicity events should be cited and discussed. For example, PMID: 37561451, 37407378, 35499382, 37691124, 35562334, 37392951.

Answer : As asked, similar studies were added to the discussion and cited.

  1. The authors are recommended to consider engaging a professional language editing service to ensure the clarity and coherence of the manuscript.

Answer : The paper has been revised for English.

Reviewer 5 Report

Comments JCM

The manuscript title “Mild cardiotoxicity and continuation of trastuzumab treatment 2 in the context of HER2-positive breast cancer” is under the scope of journal and interesting. Manuscript can be accepted after minor revision and the comments are given below.

1.        Remove the term Background, Method, results and conclusion from the abstract. Line no. 8-20

2.       There is need to update the introduction by add latest research and information in trastuzumab mediated cardiotoxicity clinically and experimentally. Authors can cite the

3.       In material Methods: Authors must mention the ethical approval and clearance along with number.

4.       In material and methods Authors should mention separate sub heading “Demographic condition and criteria”  in which mentioned the clear demographic on the map. Also mentioned the city, state name of hospital from where data has been collected. What were the demographic criteria? When the study was carried out mentioned the period?

5.       Echocardiographic Data: Which machines have used in this study clearly mentioned its model no, company and complete address of the company. Line No 105 to 117

6.       Data Collection: Mentioned clearly about source of data collection, name of hospital etc.

7.       Statistical analysis: delete the line no 135 to 139 “Matching of controls was performed using hier-135 archical clustering in JMP Pro 14 (SAS Institute, Cary NC) with five randomization factors: 136 body mass index (BMI), age, LVEF before TTZ treatment, hypertension, and diabetes. The 137 t-test or Fisher's exact test was used for post-matching verification and for comparing sub-138 groups of patients with mild and severe cardiotoxicity” keep in Demographic condition and criteria.

8.       Results section: write the complete address of the hospital line no 146

9.       Table 1. Authors should mentioned separate table for all 23 patients that include sex, Age, Body mass index and average BMI, average Age etc.

10.   Correct the BMI results  by removing comma from the value

11.   Table 1 Authors must mentioned the value represents in table as a foot note Mean± SD or SEM along with number of replicates (n=?)

12.   Table 2 authors must mention about percentage criteria.

13.   Table 3 Authors must mention the value represents in table is Mean± SD or SEM?

14.   Figure 2 is missing in the manuscript.

15.   Discussion: Authors should mentioned the mechanism of cardiotoxicity by trastuzumab kindly see these articles should cite in the manuscript.

a.       Eaton, H., Timm, K.N. Mechanisms of trastuzumab induced cardiotoxicity – is exercise a potential treatment?. Cardio-Oncology 9, 22 (2023). https://doi.org/10.1186/s40959-023-00172-3

b.       Khan G, Alam MF, Alshahrani S, Almoshari Y, Jali AM, Alqahtani S, Khalid M, Mir Najib Ullah SN, Anwer T. Trastuzumab-Mediated Cardiotoxicity and Its Preventive Intervention by Zingerone through Antioxidant and Inflammatory Pathway in Rats. Journal of Personalized Medicine. 2023; 13(5):750. https://doi.org/10.3390/jpm13050750

c.        Nemeth BT, Varga ZV, Wu WJ, Pacher P. Trastuzumab cardiotoxicity: from clinical trials to experimental studies. Br J Pharmacol. 2017 Nov;174(21):3727-3748. doi: 10.1111/bph.13643. Epub 2016 Nov 25. PMID: 27714776; PMCID: PMC5647179.

d.       Linn M et al 2022.The Research Progress of Trastuzumab-Induced Cardiotoxicity in HER-2-Positive Breast Cancer Treatment.

Author Response

The manuscript title “Mild cardiotoxicity and continuation of trastuzumab treatment 2 in the context of HER2-positive breast cancer” is under the scope of journal and interesting. Manuscript can be accepted after minor revision and the comments are given below.

Answer : We thank the reviewer for the the positive comments.

  1. Remove the term Background, Method, results and conclusion from the abstract. Line no. 8-20

Answer : As asked, the terms were removed.

  1. There is need to update the introduction by add latest research and information in trastuzumab mediated cardiotoxicity clinically and experimentally. Authors can cite the

Answer : As asked, the introduction was updated. We also updated the discussion with recent research.

  1. In material Methods: Authors must mention the ethical approval and clearance along with number.

Answer : As this is an observational retrospective study no ethical approval was included.

  1. In material and methods Authors should mention separate sub heading “Demographic condition and criteria”  in which mentioned the clear demographic on the map. Also mentioned the city, state name of hospital from where data has been collected. What were the demographic criteria? When the study was carried out mentioned the period?

Answer : We thank the reviewer for this comment. However, we did not have demographic criteria for this study apart from the control group. The demographics are presented in the result section as well as the study period. The state name of the hospital as well as the city are mentioned in the result section Line No 183 and 184.

  1. Echocardiographic Data: Which machines have used in this study clearly mentioned its model no, company and complete address of the company. Line No 105 to 117

Answer : As asked, this information was added to the manuscript.

  1. Data Collection: Mentioned clearly about source of data collection, name of hospital etc.

Answer : As asked this information was included in the article.

  1. Statistical analysis: delete the line no 135 to 139 “Matching of controls was performed using hier-135 archical clustering in JMP Pro 14 (SAS Institute, Cary NC) with five randomization factors: 136 body mass index (BMI), age, LVEF before TTZ treatment, hypertension, and diabetes. The 137 t-test or Fisher's exact test was used for post-matching verification and for comparing sub-138 groups of patients with mild and severe cardiotoxicity” keep in Demographic condition and criteria.

Answer : As answered above, a new paragraph was not added to the manuscript.

  1. Results section: write the complete address of the hospital line no 146

Answer : We thank the reviewer for his comment, however, the name of the hospital, city, province and country are described in the affiliation of the authors. Writing the complete address in the body of the article does not seem relevant for the study.

  1. Table 1. Authors should mentioned separate table for all 23 patients that include sex, Age, Body mass index and average BMI, average Age etc.

Answer : As this information is not relevant to our study, this table was not included. The mean characteristics are mentioned in the baseline characteristics.

  1. Correct the BMI results  by removing comma from the value

Answer : As asked, the coma was replaced by a point.

  1. Table 1 Authors must mentioned the value represents in table as a foot note Mean± SD or SEM along with number of replicates (n=?)

Answer : we thank the reviewer for this comment. This information was added as a footer of table 1.

  1. Table 2 authors must mention about percentage criteria.

Answer : we thank the reviewer for this comment. This information was added in table 2.

  1. Table 3 Authors must mention the value represents in table is Mean± SD or SEM?

Answer : we thank the reviewer for this comment. This information was added in table 2.

  1. Figure 2 is missing in the manuscript.

Answer : we thank the reviewer for this comment. The manuscript was adapted accordingly.

  1. Discussion: Authors should mentioned the mechanism of cardiotoxicity by trastuzumab kindly see these articles should cite in the manuscript.

Answer : We thank the reviewer for this suggestion, however, the mechanism of cardiotoxicity was described in the introduction and thus not added to the discussion.

Round 2

Reviewer 1 Report

The authors have adressed all my comments

Author Response

We thank the reviewer for his positive comment

Reviewer 2 Report

„Nonparametric tests were used” - The results should indicate where and exactly what test was used.

For such a small sample size (n < 30), the nonparametric equivalent of analysis of variance should be used.

One-way analysis of variance (ANOVA) was performed to assess the evolution of variables over the 18-month follow-up period.” - These are not different groups of patients, but one group. These are repeated measurements.

The test used should be indicated above each table in which p values are given.

Minor editing of English language required.

Author Response

Answer : As requested by the reviewer we have implemented the type of test used in the tables.The information is now available in the footnotes/title of the tables

Table 3: T-test or fisher exact test for comparisons

Figures 3,4: Fitted cubic spline

Table 5 : Dunnett post ANOVA tests

Figure 5: T-test or fisher exact test for comparisons

Appendix B: One-way ANOVA

Reviewer 4 Report

The revised manuscript has made a great improvement. I have no more comments and recommends.

Minor editing of English language required

Author Response

We thank the reviewer for his positive comment.